# Adjuvant Intravitreal Bevacizumab Injection for Choroidal and Orbital Metastases of Refractory Invasive Ductal Carcinoma of the Breast

**DOI:** 10.3390/medicina57050404

**Published:** 2021-04-22

**Authors:** I-Hung Lin, Bo-I Kuo, Fang-Yu Liu

**Affiliations:** 1Department of Ophthalmology, Tri-Service General Hospital, National Defense Medical Center, Taipei City 11490, Taiwan; petercard@gmail.com; 2Department of Ophthalmology, Taipei City Hospital Renai Branch, Taipei City 10629, Taiwan; d08421008@ntu.edu.tw; 3Department of Ophthalmology, National Taiwan University Hospital, Taipei City 10002, Taiwan

**Keywords:** bevacizumab, breast cancer, choroidal metastasis, intravitreal injection, orbital metastasis

## Abstract

The efficacy of combined intravitreal bevacizumab injection with systemic chemotherapy, palliative radiotherapy, and hormonal therapy to treat choroidal and orbital metastases is not known. Herein, we report the case of a 48-year-old woman with systemic chemotherapy-resistant choroidal and orbital metastases of the left eye originating from a stage IV invasive ductal carcinoma of the left breast. We describe the addition of a single intravitreal injection of bevacizumab in addition to treatment with systemic chemotherapy, hormonal therapy, and palliative radiotherapy. The patient’s outcome at 6-month follow-up was favorable, as the metastatic lesion reduced in size and visual acuity improved. Combined treatment with intravitreal bevacizumab injection, systemic chemotherapy, palliative radiotherapy, and hormonal therapy can resolve ocular metastatic lesions originating from breast cancers.

## 1. Introduction

Breast cancer is the most common cancer occurring in females in the world, with an incidence rate of over 1.6 million cases per year [1]. Metastatic site of breast cancer is common in bone, lung, brain, and liver [2]. However, it also can have ocular metastasis with prevalence varied from 4.6% to 9.2% [3,4]. Although systemic chemotherapy and local radiotherapy are widely used and are efficient in the control of choroidal and orbital metastatic tumors originating from breast cancer, decreased vision and intolerable ocular pain may still occur and progress [5]. Bevacizumab is a full-length recombinant humanized monoclonal antibody against all forms of vascular endothelial growth factor A, which can be used as a target therapy for the treatment of metastatic breast cancer [6]. Intravitreal injection of bevacizumab, combined with systemic chemotherapy, hormone therapy, palliative radiotherapy, transpupillary thermotherapy, and/or photodynamic therapy, may improve the visual outcome. A few recent case reports have described the management of metastatic choroidal lesions from breast cancer through intravitreal injection of bevacizumab combined with the above therapies with encouraging results [5,7,8,9,10,11,12,13,14]. Herein, we report the case of a 48-year-old woman with stage IV invasive ductal carcinoma of her left breast, who presented with choroidal and orbital metastases of the left eye that were refractory to systemic chemotherapy. We describe this patient’s treatment with a single intravitreal injection of bevacizumab in combination with systemic chemotherapy, hormone therapy, and palliative radiotherapy and reveal the outcomes at the 6-month follow-up.

## 2. Case Report

In 2015, a 48-year-old Chinese woman with stage II estrogen-sensitive (estrogen receptor-positive, progesterone receptor-positive, human epidermal growth factor receptor 2 (HER2) equivocal) invasive ductal carcinoma of the left breast was treated with mastectomy of her left breast, eight rounds of chemotherapy, and 30 rounds of radiotherapy in China. Two years later, she was found to have bone, liver, and lung metastases; her cancer staging was modified to stage IV, and she received four additional rounds of chemotherapy in China. About 3 years after the mastectomy, she was noted to have rapidly decreasing vision in both eyes. She was diagnosed with metastasis to both eyes and received three courses of chemotherapy with ramucirumab, vinorelbine, and fluorouracil in another hospital in Taiwan. Due to persistent blurred vision, she presented to our hospital for further management at about three years and two months after the mastectomy.

At that time, best-corrected visual acuity (BCVA) was no light perception (LP) in the right eye and 20/120 in the left eye. Visual evoked potential (VEP) exam showed flat wave in the right eye and delay of P100 wave in the left eye. Intraocular pressure was 8 mmHg in the right eye and 13 mmHg in the left eye. Both anterior segments were unremarkable except that the pupils dilated with no light reflex in the right eye. Fundoscopy revealed choroidal detachment (CD) and exudative macular-off retinal detachment (RD) of the inferior side at 4 to 10 o’clock in the right eye. In the left eye, multiple white-yellowish spot lesions were seen at the fovea with multiple irregular, ill-defined white lesions at the 2 to 3 o’clock position on the periphery. Optical coherence tomography (OCT) revealed poor signal in the right eye and a subretinal elevated solid lesion at the fovea, with subretinal fluid collection at 2 to 3 o’clock on the peripheral side in the left eye. B-mode ultrasonography revealed kissing choroidals with RD in the right eye and a choroidal mass with an extension of 7.4 × 3.8 mm on the superior temporal side in the left eye (Figure 1).

T2-weighted magnetic resonance imaging (MRI) demonstrated intraocular, orbital, and optic nerve enhancement in the right eye and a choroidal mass at the temporal side in the left eye with orbital rim enhancement (Figure 2). Our diagnoses were intraocular, orbital, and optic nerve metastases of the right eye, as well as choroidal and orbital metastases of the left eye secondary to invasive ductal carcinoma of the left breast.

The patient received one 2.5 mg/0.1 mL bevacizumab (Avastin**^®^**, Roche, Inc., Basel, Switzerland) intravitreal injection in the left eye. Two courses of concurrent systemic chemotherapy with Avastin, vinorelbine (Navelbine**^®^**, Pierre Fabre Medicament, Paris, France), and cisplatin (Kemoplat**^®^**, Fresenius Kabi Oncology Limited, Haryana, India) were administered over 2 weeks, and palliative radiotherapy to the bilateral orbital areas with sparing of the anterior chambers was administered in 10 fractions at a dosage of 3 Gy per fraction over 10 days.

Six weeks after intravitreal injection of bevacizumab, the BCVA of the left eye improved to 20/30. Fundal examination of the left eye revealed that the multiple white-yellowish spot lesions at the fovea had almost disappeared, and the multiple irregular, ill-defined white lesions at 2 to 3 o’clock on the peripheral side had diminished in size. OCT revealed that the subretinal elevated solid lesion had regressed at the fovea and the subretinal fluid collection at 2 to 3 o’clock on the peripheral side in the left eye had also diminished. B-mode ultrasonography revealed a dramatic decrease in the size of the tumor, with an extension of 2.7 × 0.7 mm on the superior temporal side in the left eye (Figure 3). For the right eye, the BCVA, fundal examination, OCT, and B-mode ultrasonography findings were stationary. No ocular or systemic complications were observed.

Next, the patient received oral letrozole (Femara**^®^**, Novartis Pharma Stein AG, Basel, Switzerland) hormone therapy at a dosage of 2.5 mg daily for 1 month, and then intravenous (IV) fulvestrant (Faslodex**^®^**, AstraZeneca UK Limited, Cheshire, UK) hormone therapy at a dosage of 250 mg once every 3 weeks for three cycles. Concurrent ribociclib (Kisqali**^®^**, Novartis Pharmaceuticals Corporation, East Hanover, NJ, USA) chemotherapy was administered orally at a dosage of 600 mg daily for 21 consecutive days followed by 7 days off treatment, for 4 months.

Six months after the intravitreal injection of bevacizumab, the BCVA of the left eye had improved to 20/20. Fundal examination and OCT of the left eye revealed stable findings. B-mode ultrasonography of the left eye showed that the tumor had almost disappeared, and its size was too small for measurement (Figure 4).

In the right eye, the BCVA improved to LP. The fundal examination, OCT, and B-mode ultrasonography revealed regression of both CD and exudative RD in the right eye. No ocular or systemic complications were observed at the end of follow-up.

This case study was approved by the institutional ethics committee and the patient described in this study provided informed consent for the publication of this case report. All procedures in this study adhered to the tenets of the Declaration of Helsinki.

## 3. Discussion

This case report showed that a combination of intravitreal bevacizumab with systemic chemotherapy, palliative radiotherapy, and hormone therapy can be an effective, rapid, and safe treatment option for patients with choroidal and orbital metastases of invasive ductal carcinoma of the breast.

The choroid, which has a rich supply of blood vessels, is the most frequent site of intraocular metastasis, originating most commonly from a primary cancer of the breast (47%) or the lung (21%) [5,15]. Metastases to the orbit also frequently originate from a primary cancer of the breast (53%) [16]. Both the choroidal and orbital metastasis have poor prognosis, with mean survival after diagnosis of such metastases is 13 months in choroid metastasis [17] and 31 months in orbital metastasis [18].

Although systemic chemotherapy and local radiotherapy are often used for the control of choroidal and orbital metastatic tumors, decreased vision and intolerable ocular pain may still occur and progress [5]. Further, while up to 89% of patients with orbital metastases may respond to radiotherapy with tumor regression and improvement of symptoms, therapy-induced complications like cataracts, exposure keratopathy, iris neovascularization, radiation retinopathy, and papillopathy have been reported in 12% of cases [19,20]. Adjuvant treatments, like intravitreal injection of bevacizumab, may therefore be necessary.

Bevacizumab is an anti-vascular endothelial growth factor therapy that may have anti-angiogenic and anti-permeability effects on new tumor vessels, inhibiting the growth of tumor cells [7]. It can be used as a systemic targeted therapy by intravenous infusion for breast cancer with or without metastasis [8]. However, intravenous infusion may not provide good ocular penetration and enough drug concentration for the treatment of ocular metastasis. Intravitreal injection of bevacizumab can be a solution to increase the intraocular drug concentration and the treatment effect [21].

There have been 11 reported cases of patients with choroidal lesions secondary to breast cancer that have been successfully treated with intravitreal bevacizumab despite failing systemic treatments [5,7,8,9,10,11,12,13,14]. Most of these cases had received intravitreal bevacizumab combined with chemotherapy, palliative orbital radiotherapy, or hormone therapy, with one combined with transpupillary thermotherapy [7] and one combined with indocyanine green-mediated photothrombosis [13]. Failure of intravitreal bevacizumab in the treatment of choroidal metastasis secondary to breast cancer was also reported in one case [12]. However, in these prior case reports, there were isolated choroidal metastases without orbital metastases.

Different from the aforementioned reports, our patient had both choroidal and orbital metastases that did not improve after three courses of systemic chemotherapy with ramucirumab, vinorelbine, and fluorouracil. Circumferential large choroidal detachment with concurrent retinal detachment in right eye may increase the complication of intravitreal injection. Poor visual acuity and flat VEP made the recovery in the right eye more unfavorable. Therefore, after discussing details with the patient, we decided to apply intravitreal injection of bevacizumab in the left eye. Six weeks after the addition of intravitreal bevacizumab treatment to systemic chemotherapy and palliative orbital radiotherapy, visual acuity had improved, choroidal tumor size had decreased, and the subretinal fluid collection had resolved in the left eye. This treatment effect was sustained for 6 months with a combination of hormone therapy and chemotherapy. No ocular or systemic complications were observed during follow-up.

## 4. Conclusions

A combination of intravitreal bevacizumab with systemic chemotherapy, palliative radiotherapy, and hormone therapy can be an effective, rapid, and safe treatment option in patients with choroidal and orbital metastases of invasive ductal carcinoma of the breast. It can reduce subretinal fluid and tumor size, improving visual acuity and the quality of life of patients.

## Figures and Tables

**Figure 1 medicina-57-00404-f001:**
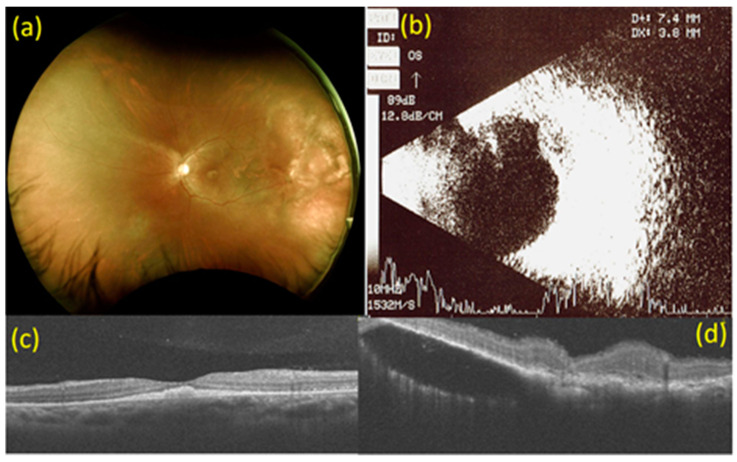
Images at baseline. (**a**) Fundoscopy at baseline. Multiple white-yellowish spot lesions at the fovea with multiple irregular, ill-defined white lesions at 2 to 3 o’clock on the peripheral side in the left eye. (**b**) B-mode ultrasonography at baseline. A choroidal mass with an extension of 7.4 × 3.8 mm on the superior temporal side in the left eye. (**c**) Macular optical coherence tomography (OCT) scan at baseline. Subretinal elevated solid lesion at the fovea in the left eye. (**d**) OCT scan of choroidal mass at 2 to 3 o’clock on the peripheral side in the left eye at baseline. Subretinal fluid collection at 2 to 3 o’clock on the peripheral side in the left eye.

**Figure 2 medicina-57-00404-f002:**
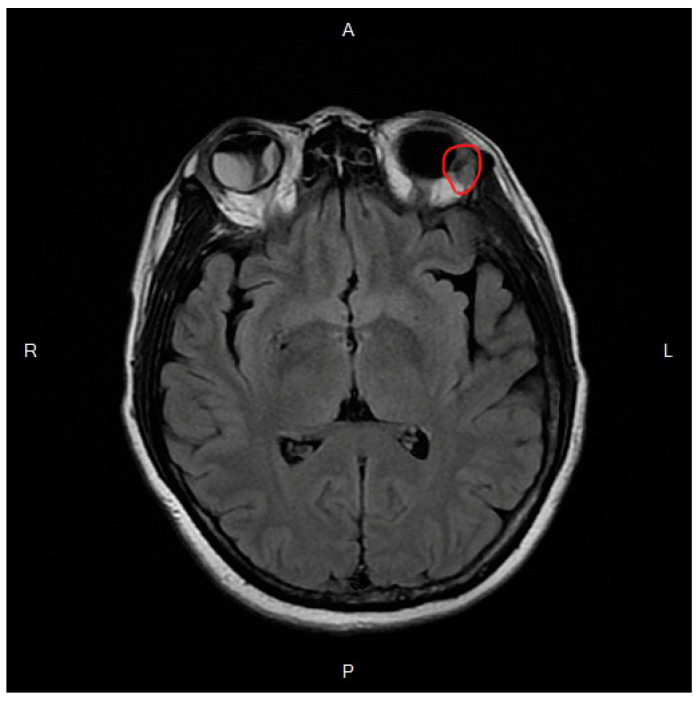
T2-weighted magnetic resonance image demonstrates retinal detachment in the right eye and a lesion on the temporal side with orbital rim enhancement in the left eye (red circle).

**Figure 3 medicina-57-00404-f003:**
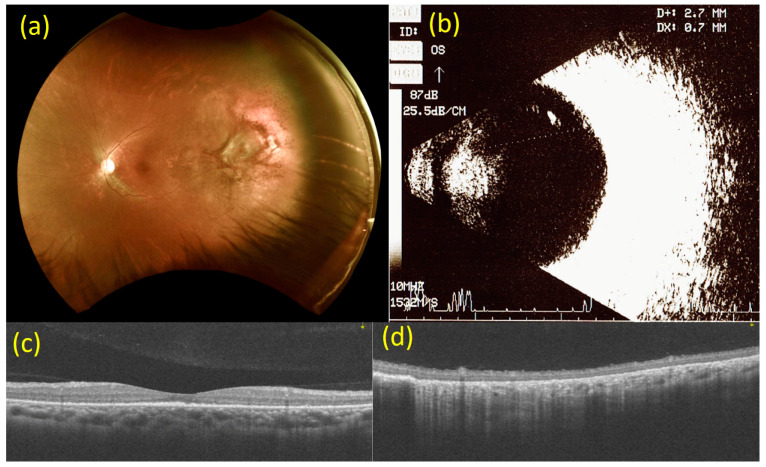
Images from 6 weeks after injection of bevacizumab (2.5 mg). (**a**) Fundoscopy 6 weeks after the injection of bevacizumab (2.5 mg). The white-yellowish spot lesions at the fovea have almost disappeared, and the multiple irregular, ill-defined white lesions at 2 to 3 o’clock on the peripheral side in the left eye have diminished in size. (**b**) B-mode ultrasonography 6 weeks after the injection of bevacizumab (2.5 mg). A dramatic decrease in the size of the tumor, with an extension of 2.7 × 0.7 mm at the superior temporal side in the left eye. (**c**) Macular optical coherence tomography (OCT) scan 6 weeks after the injection of bevacizumab (2.5 mg). The subretinal elevated solid lesion has regressed at the fovea in the left eye. (**d**) OCT scan of the choroidal mass at 2 to 3 o’clock on the peripheral side 6 weeks after the injection of bevacizumab (2.5 mg). The subretinal fluid collection has regressed at the 2 to 3 o’clock position on the peripheral side in the left eye.

**Figure 4 medicina-57-00404-f004:**
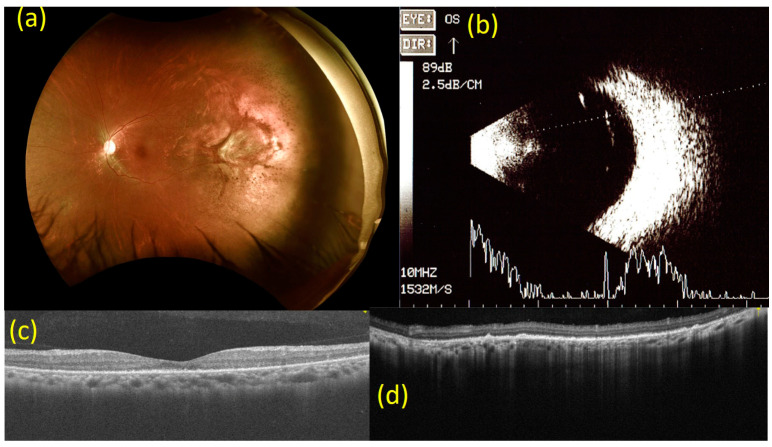
Images from 6 months after the injection of bevacizumab (2.5 mg). (**a**) Fundoscopy 6 months after the injection of bevacizumab (2.5 mg). Findings are consistent with imaging done at 6-week follow-up. (**b**) B-mode ultrasonography 6 months after the injection of bevacizumab (2.5 mg). The tumor in the left eye has almost disappeared and its size is too small to quantify. (**c**) Macular optical coherence tomography (OCT) scan 6 months after the injection of bevacizumab (2.5 mg). Findings are consistent with imaging done at a 6-week follow-up for the left eye. (**d**) OCT scan of the choroidal mass at the 2 to 3 o’clock position on the peripheral side in the left eye 6 months after the injection of bevacizumab (2.5 mg). Findings are consistent with the imaging conducted at 6-week follow-up.

## Data Availability

The data used and/or analyzed during the current study are not available for public access because of patient privacy concerns but are available from the corresponding author on reasonable request.

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
