# Peer review of "Adjuvant Intravitreal Bevacizumab Injection for Choroidal and Orbital Metastases of Refractory Invasive Ductal Carcinoma of the Breast"

_medicina, 2021, doi:10.3390/medicina57050404_

Round 1
Reviewer 1 Report
The aim of the paper is to report a case of a patient with choroidal and orbital metastasis, which is rare in clinical practice.
Critical notes:
Introduction does not provide sufficient background and does not include any relevant references on the topic.
It is not clear why authors decided to treat only left eye of the patient. It is interesting to share their arguments.
It is written that “ The patient received one 0.1 mL bevacizumab…”, but the reader does not understand how many milligrams bevacizumab was used. It is important to be clear, because there are different options: 1.25mg, 2,5mg…
It is written “There have only been six reported cases of patients with choroidal lesions secondary to breast cancer that have been successfully treated with intravitreal 162 bevacizumab despite failing systemic treatments [1,5,6,7–9]”. In my opinion reported cases are more.
There are more recent publications on this topic (from last 5 years) that are not discussed and cited by authors.
Author Response
Response to Reviewer 1 Comments
Point 1: Introduction does not provide sufficient background and does not include any relevant references on the topic.
Response 1: We revised it carefully with sufficient background and relevant references at line 26-39. Thanks for kindly suggestion.
Point 2: It is not clear why authors decided to treat only left eye of the patient. It is interesting to share their arguments.
Response 2: First of all, circumferential large choroidal detachment with concurrent retinal detachment in right eye may increase the complication of intravitreal injection. Second, the poor visual acuity and flat wave of visual evoked potential (VEP) exam made the recovery in right eye more unfavorable (Figure 1). Third, the use of bevacizumab intravitreally is currently off-label use without health insurance support in Taiwan which may cause a financial burden to the patient. Therefore, after detail discussing with patient, we decided to apply intravitreal injection of bevacizumab in left eye. We also clarify it in the manuscript at line 60-61 and 193-197.
Figure 1 Visual evoked potential (VEP) exam showed flat wave in the right eye (left column) and delay of P100 wave in the left eye (right column).
Point 3: It is written that “The patient received one 0.1 mL bevacizumab…”, but the reader does not understand how many milligrams bevacizumab was used. It is important to be clear, because there are different options: 1.25mg, 2,5mg…
Response 3: It was 2.5mg/0.1 mL bevacizumab and we revised it in the manuscript at line 90. Thanks for the suggestion.
Point 4: It is written “There have only been six reported cases of patients with choroidal lesions secondary to breast cancer that have been successfully treated with intravitreal 162 bevacizumab despite failing systemic treatments [1,5,6,7–9]”. In my opinion reported cases are more.
There are more recent publications on this topic (from last 5 years) that are not discussed and cited by authors.
Response 4: We revised it with more new cases and publications in the manuscript at line 181-189. Thanks a lot.
Reviewer 2 Report
I think is a novel and well-written paper, but I would like some parts of it to be clarified.
The introduction should be expanded and indicate why the presentation of this case is relevant. Why is only one eye treated? The discussion should be expanded
Author Response
Response to Reviewer 2 Comments
Point 1: The introduction should be expanded and indicate why the presentation of this case is relevant.
Response 1: We revised it with sufficient background and relevant references in the manuscript at line 26-39. Thanks for the kind suggestion
Point 2: Why is only one eye treated?
Response 2: First of all, circumferential large choroidal detachment with concurrent retinal detachment in right eye may increase the complication of intravitreal injection. Second, the poor visual acuity and flat wave of visual evoked potential (VEP) exam made the recovery in right eye more unfavorable (Figure 1). Third, the use of bevacizumab intravitreally is currently off-label use without health insurance support in Taiwan which may cause a financial burden to the patient. Therefore, after detail discussing with patient, we decided to apply intravitreal injection of bevacizumab in left eye. We also clarify it in the manuscript at line 60-61 and 193-197.
Figure 1 Visual evoked potential (VEP) exam showed flat wave in the right eye (left column) and delay of P100 wave in the left eye (right column).
Point 3: The discussion should be expanded
Response 3: We expand the discussion with following information: the prognosis of choroid metastasis and orbital metastasis, discussion about the different drug deliver way of bevacizumab, more new cases and publications, and why we treated left eye only in the manuscript at line 161-163, 176-180, 181-189 and 193-197. Thanks for carefully review and kind suggestion.
Round 2
Reviewer 1 Report
The manuscript is better and more useful for readers now.
This manuscript is a resubmission of an earlier submission. The following is a list of the peer review reports and author responses from that submission.